# Continuous HIV-1 Escape from Autologous Neutralization and Development of Cross-Reactive Antibody Responses Characterizes Slow Disease Progression of Children

**DOI:** 10.3390/vaccines9030260

**Published:** 2021-03-14

**Authors:** Stefania Dispinseri, Mariangela Cavarelli, Monica Tolazzi, Anna Maria Plebani, Marianne Jansson, Gabriella Scarlatti

**Affiliations:** 1Viral Evolution and Transmission Unit, Division of Immunology, Transplantation and Infectious Diseases, IRCCS Ospedale San Raffaele, 20132 Milan, Italy; dispinseri.stefania@hsr.it (S.D.); tolazzi.monica@hsr.it (M.T.); 2Inserm, CEA, Center for Immunology of Viral, Auto-Immune, Hematological and Bacterial Diseases (IMVA-HB/IDMIT), University Paris-Saclay, 92265 Fontenay-aux-Roses & Le Kremlin-Bicêtre, France; mariangela.cavarelli@cea.fr; 3Pediatric Emergency Unit, Filippo Del Ponte Hospital, ASST-Settelaghi, 21100 Varese, Italy; annample@yahoo.it; 4Department of Laboratory Medicine, Lund University, 22242 Lund, Sweden; marianne.jansson@med.lu.se

**Keywords:** HIV-1, neutralization, ADCC, children, humoral immunity, disease progression

## Abstract

The antibodies with different effector functions evoked by Human Immunodeficiency Virus type 1 (HIV-1) transmitted from mother to child, and their role in the pathogenesis of infected children remain unresolved. So, too, the kinetics and breadth of these responses remain to be clearly defined, compared to those developing in adults. Here, we studied the kinetics of the autologous and heterologous neutralizing antibody (Nab) responses, in addition to antibody-dependent cellular cytotoxicity (ADCC), in HIV-1 infected children with different disease progression rates followed from close after birth and five years on. Autologous and heterologous neutralization were determined by Peripheral blood mononuclear cells (PBMC)- and TZMbl-based assays, and ADCC was assessed with the GranToxiLux assay. The reactivity to an immunodominant HIV-1 gp41 epitope, and childhood vaccine antigens, was assessed by ELISA. Newborns displayed antibodies directed towards the HIV-1 gp41 epitope. However, antibodies neutralizing the transmitted virus were undetectable. Nabs directed against the transmitted virus developed usually within 12 months of age in children with slow progression, but rarely in rapid progressors. Thereafter, autologous Nabs persisted throughout the follow-up of the slow progressors and induced a continuous emergence of escape variants. Heterologous cross-Nabs were detected within two years, but their subsequent increase in potency and breadth was mainly a trait of slow progressors. Analogously, titers of antibodies mediating ADCC to gp120 BaL pulsed target cells increased in slow progressors during follow-up. The kinetics of antibody responses to the immunodominant viral antigen and the vaccine antigens were sustained and independent of disease progression. Persistent autologous Nabs triggering viral escape and an increase in the breadth and potency of cross-Nabs are exclusive to HIV-1 infected slowly progressing children.

## 1. Introduction

The rational design of an effective vaccine against Human Immunodeficiency Virus type 1 (HIV-1) requires an understanding the functional characteristics of antibodies capable of preventing transmission of the virus or providing a benefit to the disease in terms of severity of symptoms and/or progression to a fatal outcome. The ideal vaccine should be able to evoke cross- neutralizing antibodies (Nabs) to prevent transmission of HIV-1 but other antibody effector functions such as antibody-dependent cellular cytotoxicity (ADCC) have been suggested to protect from HIV as well [1]. In specific, during mother-to-child transmission (MTCT) of HIV-1 different specificities and effector function of the antibodies may impact the risk of transmission according to the route of infection, i.e., during pregnancy, at delivery or via breast feeding. In line with this, it has to be considered that HIV-1 infection of children born to infected mothers has some specific features compared to infection of adults. Indeed, while the majority of children develop AIDS slowly over several years, in contrast to adults approximately one-quarter of them has a rapidly progressing disease, and develops features characteristic of AIDS within the first year of life [2,3,4]. Knowledge on the immunological mechanisms underlying the different patterns of disease progression in HIV-1 infected children is still lacking.

In HIV-1 infected adults Nabs against the autologous virus emerge within weeks from infection [5,6,7], but usually the virus readily escapes this response, possibly due to their narrow specificity. Still, 10–30% of HIV-1 infected individuals develop within two to four years from infection antibodies cross-reactive with viruses isolated from other infected individuals [8] and of different subtypes [9]. However, these cross-Nab responses are not necessarily associated with delayed disease progression in infected adults [10], suggesting that other virus controlling immune responses may play more essential roles in the determination of disease progression rates.

Most of published studies on the Nab responses in infants are cross-sectional [11,12,13], however, some more recent studies have suggested that HIV-infected children are able to develop broader and more potent virus neutralization earlier than adults and via a distinct mechanistic pathway, highlighting potential advantages of the child’s immune system in eliciting broad Nabs (bNabs) compared to adults [14,15,16,17]. These papers proposed that exposure to high antigen concentrations at high CD4+ T cell count as described in children, or the robust ability to mount responses mediated by T helper cells, may contribute to the development of bNAbs. In addition, passively acquired maternal antibodies mediating ADCC were significantly associated with improved survival of infected infants and with improved infant outcomes and reduced set point viral load [11,18,19,20].

It is still a matter of discussion if maternal Nabs may assert a selection pressure on the transmitted virus variant, favoring transmission of escape mutants, which in turn may have replication advantages in its new host. Even though the frequency, breadth, and potency of Nabs responses of non-transmitting mothers in general are better than those of transmitting ones [21], it is debated if the role of antibodies, whether Nabs or antibodies mediating other functions, may change according to the route of transmission [22,23,24,25].

In this regard, it was clearly shown that cocktails of cross-neutralizing monoclonal antibodies (mAb) directed to HIV-1 completely block acquisition of simian immunodeficiency virus expressing the HIV envelope (SHIV) in juvenile non-human primates when administrated before or even after oral administration of the virus [26,27], thus supporting the use of mAbs for prevention of infection. The recent results of the Antibody Mediated Prevention (AMP) trial, which tested the efficacy of the mAb VRC01 in preventing HIV-infection in adults, suggest that a combination of mAbs may be preferable to administer in adults. However, MTCT is characterized by a virus bottleneck in which the child’s infection is established by a single or a few transmitted viral variants [28,29,30].

Indeed, antibody specificities may contribute to the transmission bottleneck in that it was shown that none of the antibodies obtained from mothers close to delivery were able to neutralize the transmitted variant [31]. Interestingly, several studies showed that infant’s early viruses are more resistant to the b12, 2G12, 2F5, and 4E10 broadly neutralizing mAbs than the maternal viruses, but are as sensitive to antibodies directed against quaternary and CD4-binding site epitopes [22,30,32,33,34]. Thus, infant’s early transmitted/founder viruses may raise Nab responses, which may target selected regions of the envelope glycoproteins (Env), and thus be relevant for vaccine development. 

A peculiarity of HIV-1 infected newborns is that they have antiviral maternal antibodies acquired through trans-placental passage, which may impact the onset and the specificity of their antibody responses. It is known that in HIV-1 uninfected newborns Nabs normally decay within 6 months of age [35]. However, in infected newborns, maternal Nabs are rarely detected [23,24,32,35,36,37], possibly because transmitting mothers often lack Nabs targeting the transmitted virus. 

Thus, to better clarify the role of the antibody responses in children with different HIV-1 disease progression, we studied the development of functional antibody responses, Nabs against autologous and heterologous viruses and ADCC, as well as of antibody reactivities to HIV-1 Env and childhood vaccine antigens. We show that slowly, but seldom rapidly, progressing children develop autologous Nabs. Increasing neutralizing titers are detected mainly against the virus isolated early after transmission, and escape from neutralization is a common event. Interestingly, we also demonstrate that heterologous cross-Nabs are already detected within two years of age, but the increase in potency and breadth thereafter is limited to slow progressors.

## 2. Material and Methods

### 2.1. Patients’ Characteristics and Samples

The 25 HIV-1 infected children included in this study were from a large prospective study on MTCT conducted between 1989 and 1994 [38]. None of the children was breastfed, whereas data were insufficient to define in utero vs. intrapartum transmission route. Children were all infected with a subtype B virus as previously shown [39,40]. At the time of the cohort study plasma HIV-1 p24 antigen (Ag) or RNA load were not routinely determined and thus, not available. Clinical and immunological staging was according to the classification system of the Centers for Disease Control (CDC) [41]. Children who experienced a severe decline of the CD4+ T cells entering CDC immunological category 3 or died within two years were defined rapid progressors (Table 1), whereas those classified as CDC 3 after two years or who did not enter this category by at least five years of follow up were defined slow progressors. Treated children, as indicated in Table 1, received only mono or dual antiretroviral therapy (ART) according to the national guidelines at the time of the prospective MTCT study.

HIV-1 isolates, obtained from patient’s peripheral blood mononuclear cells (PBMC) after coculture with mitogen stimulated donor PBMC, as previously described [38], were available from 15 children (Table 1). Determination of co-receptor usage of each isolate was performed using the U87 indicator cells expressing the chemokine receptors, either CCR5 or CXCR4 [42].

Sequential plasma samples (between two and 16 samples/child) from 15 children were tested for autologous Nabs in a PBMC-based assay against up to five primary virus isolates. While for further analysis of heterologous neutralization and ADCC, one plasma sample obtained during the first two years of age (called early plasma) and/or during years three to five (called late plasma) for each child were used. Specifically, late plasma of ten children (four rapid and six slow progressor; age range 24 to 53 months) were used in a PBMC-based neutralization assay against four heterologous primary isolates. Furthermore, an early plasma sample (age range 9–24 months; mean: 16.9 months), and/or a late sample (age range 25 to 55 months; mean: 42.5 months) from 22 children (six rapid and 16 slow progressors) were evaluated in the TZMbl-assay against three pseudotyped viruses (PSV). ADCC was performed with plasma of five rapid and eight slow progressor. Tests performed for each child are detailed in Appendix A. Plasma was inactivated 60 min at 56 °C prior to testing.

### 2.2. Ethical Permission

Ethical approval to use plasma and viral isolates to study the immunological response of HIV-1 infected children was obtained from the Ethical Committee of the San Raffaele Scientific Institute according to national laws and registered with protocol number 03072008v12.

### 2.3. PBMC-Based Neutralization Assay

In total, 47 primary virus isolates obtained from 15 children were used for the PBMC-based neutralization assays [43]. In brief, six steps of 2-fold dilution, starting with 1/20 of each heat-inactivated plasma, were incubated with 20 and 40 TCID50 of viral supernatant for 1 h in a round-bottom microtiter plate (Nunc, Roskild, Denmark). Subsequently, 10^5^ PHA-stimulated PBMC were added and plates further incubated. At days 1 and 3 cells were washed, fresh IL-2 RPMI medium supplemented with 10% FCS (Lonza group, Basel, Switzerland) added, and at day 7 each well was tested for the presence of p24 Ag in the in-house ELISA (www.aaltobioreagents.ie, accessed on 15 February 2021) (protocol available since 2007). Positive control of viral growth consisted in six wells with virus for each TCID50 and cells, whereas negative control consisted in four wells with only cells, representing the background. Neutralization titers are the reciprocal of the highest plasma dilution giving at least a 90% reduction of HIV-1 p24 Ag compared to virus control. The 90% inhibitory serum dilution (ID90) was calculated with a linear interpolation method using the mean of the duplicate responses [44].

### 2.4. TZMbl Neutralization Assay

Viral titration and neutralization assays were performed with two R5 subtype B PSVs (SF162, AC10) and one R5 subtype A PSV (VI191) [45]. AC10 and VI191 are classified as neutralization sensitivity tier 2 PSVs. Briefly, four steps of 3-fold dilution, starting with 1/20 of each heat-inactivated plasma was incubated with viral supernatant (200 TCID50) for 1 h in a flat-bottom microtiter plate (Nunc, Roskild, Denmark). Thereafter, 10^4^ TZMbl cells were added and plates further incubated for 48 h, when luciferase activity was determined with a luciferase assay system (Bright-Glo, Promega, Madison, WI, USA) and measured in a Mitras luminometer (Berthold, Bad Wildbad, Germany). Positive control of viral growth consisted in four wells with virus and cells, whereas negative control consisted in four wells with only cells, representing the background. Neutralization titers are the sample dilution at which relative luminescence units (RLU) were reduced by 50% compared to the mean of RLU in virus control wells after subtraction of background RLU in control wells with only cells. The 50% inhibitory serum dilution (ID50) was calculated with a linear interpolation method using the mean of the duplicate responses [46]. 

### 2.5. Antibody-Dependent Cellular Cytotoxicity Assay

Patients’ plasma was tested in the ADCC GranToxiLux (ADCC-GTL) assay according to the protocol described in [47]. Briefly, CEM.NKR.CCR5 cells were coated with HIV-1 subtype B gp120 of the BaL strain and incubated at 1:10 ratio with Natural Killer (NK) cells positively isolated (CD56 microbeads human, Miltenyi Biotec, Germany) from healthy donor buffy-coat and eight 5-fold dilutions of each plasma starting with 1/250 dilution. Samples were acquired at LSR Fortessa (BD Biosciences) and analyzed with FlowJo 8.8.3 (Treestar Inc., Ashland, OR, USA). Positive controls were HIV positive plasma with known ADCC capacity. Negative control was plasma from a healthy donor. The percentage of the cells positive for the Granzyme B (GzB) substrate were reported as percentage of GzB activity. The results were considered positive if the %GzB after background subtraction was >8%. Results are expressed as maximal %GzB or as linear titer, which corresponds to the reciprocal of the last serum dilution giving a positivity %GzB as calculated by determining the area under the curve (AUC) with GraphPad Prism Version 8.

### 2.6. ELISA Assay for Reactivity to Tetanus and Diphteria Toxoids, and HIV-1 Gp4

JB-4 (Ferring AB, Malmo, Sweden), an 18aa long peptide representing the immunodominant constant region of HIV-1 LAI gp41 (594–613), and tetanus or diphtheria toxoid antigens (Statens Serum Institut, Copenhagen, Denmark) were coated in microtiter plates (Maxisorb, Nunc, Denmark) at a concentration of 1 µg per well. Plasma was analyzed in six 2-fold steps starting from 1/50 in 100 μL of 0.05 M sodium carbonate buffer pH 9.6. Plates were washed five times with 0.9% NaCl containing 0.05% Tween 20 (Fluka Chemie AG, Buchs, Switzerland) before addition of sera and between each following step. Plates were first incubated with 100 μL of serum diluted from 1/100 in 2-fold steps with H-buffer (2% goat serum, 0.5% bovine serum albumin, 0.05% Tween 20 in phosphate buffered saline) for 60 min, then with goat antihuman immunoglobulin G, conjugated with alkaline phosphatase (Sigma Chemicals, St Louis, MI, USA) at a dilution of 1/1500 for 30 min, and finally with the substrate p-nitrophenyl phosphate (Sigma) for 30 min. All incubations were performed at 37 °C. The reaction was stopped with 1 M NaOH (Sigma Chemicals, St Louis, MI, USA) and the optical density was measured at 405 nm. Plasma from normal healthy blood donors were used as negative controls and a positive reaction was defined as an absorbance value of 7 SD above the mean negative control. 

### 2.7. Statistical Analysis

Neutralization titers below 1/20 were assigned a value of 0. When the end-point neutralization was not reached within the last plasma dilution used, the following dilution was considered for statistics. 

Fisher exact test was applied to compare the frequency of autologous neutralization. Wilcoxon test was applied to compare the heterologous Nabs in TZMbl-assay with early and late plasma in rapid or slow progressors. Kruskal–Wallis test was applied to perform multiple comparisons. Mann-Whitney *U* test was used to compare unpaired data. Software used for calculation was GraphPad Prism Version 8 (San Diego, CA, USA).

## 3. Results

### 3.1. Persistent Autologous Neutralizing Activity Is Common in Slow Progressing Children

To analyze the development of Nab responses in relation to disease progression, we tested the autologous neutralizing activity of plasma from 15 infected children, seven rapid and eight slow progressors, against their own viruses obtained throughout disease progression, using a PBMC-based assay. At birth no autologous Nabs were detected (Figure 1) in the six newborns, for whom a sample at birth was available (rapid progressor B196, B204, B224, and slow progressor B145, B190, B199).

At follow-up, ten children developed Nabs against one or more autologous isolate at any given time when tested in PBMC based-neutralization assays (Figure 1 and Appendix A). In six children (rapid progressors B224 and B380, and slow progressors B3, B115, B190 and B199) sampled before one year of age Nabs were already detected though at varying titers (range: 1/21–640), while in other four slow progressor (B32, B136, B145, B306) Nabs were detected thereafter. The presence of an autologous Nab response correlated with disease progression, as Nabs were detected in all eight slow progressors, but only two of seven rapid progressors (B224 and B380) (*p* = 0.0070, Fishers’ exact test). 

In general, Nabs developed against the early virus isolate and showed increasing titers and persistence throughout follow-up. While Nabs against viruses isolated later during follow-up were detected in eight children (six slow progressors: B3, 32, 136, 145, 190, 306, and two rapid progressors: B224 and B380). Contemporaneous Nabs, i.e., Nabs against a simultaneously isolated virus, were only detected at sporadic time points in five children, including the two rapid progressors (at titers 1/20–1/320). Thus, providing evidence of continuous emergence of escape variants. Three slow progressors (children B3, B136, B145) raised Nabs also to their CXCR4-using virus, which were isolated from the children during their follow-up. Thus, our results show that the development and persistence of autologous Nabs are common in slow progressors and rare in rapid progressors. Of note, the two rapid progressors who developed Nabs were the only still alive after four years of age.

### 3.2. Neutralization against Heterologous Virus Evolves in Children with Slow Disease Progression

In order to analyze the development of heterologous cross-Nab responses the late plasma of ten children was tested in the PBMC-based assay against four heterologous primary viruses isolated from three slow progressing children (Table 2). Plasma of one of four rapid and three of six slow progressors showed low titer Nab activity (titers 1/20–80) against one or two of the viruses tested. Two slow progressors neutralized also the X4 virus isolate (number 3–54). Thus, heterologous neutralization against primary isolates is sporadic when assessed with a PBMC-based assay.

An extended analysis of heterologous cross-Nab responses was performed with early and late plasma of six rapid and 16 slow progressors using the TZMbl-assay. All children developed, at least at one time point, Nabs against Tier 1 subtype B PSV SF162 with varying titers (range: 1/20–1/1620) (Figure 2). The Tier 2 PSV AC10 (subtype B) was neutralized by 64% of early and 89.5% of late plasma, and PSV VI191 (subtype A) by 54% of early and 84% of late plasma.

In those children, for whom paired samples were tested, titers of heterologous Nabs increased at least of 2-fold in six of eight slow progressors against all three viruses, whereas in rapid progressors titers usually decreased (Appendix A). If we applied more stringent criteria, as elsewhere suggested [9], considering a 1/100 plasma dilution as cut-off, still late plasma of four slow progressor would neutralize all three PSVs but none of the rapid progressor. These results demonstrate that slow disease progression is associated with the development, enlargement of breadth and increase of titers over time of a Nab response against heterologous viruses.

### 3.3. Titers of Antibodies Mediating ADCC against gp120 Coated Target Cells Increase with Age

The capacity of plasma to mediate ADCC against BaL gp120 coated target cells was tested in eight slow and five rapid progressors with the same early and/or late plasma used to detect neutralization. All except one (B3) slow progressing children showed ADCC activity at the early time point (mean titer 1/34,680), which increased in titer in all children at the late serum sample (mean titer 1/90,917) (Figure 3). In children with paired samples the increase ranged from 3.5 to 84 times. The maximal percent GzB release increased or remained stable in all children with a mean 31% at the early sample and 52% at the late sample tested. Two of the rapid progressors, showed an ADCC activity at the early timepoint, which decreased in one child (B380) and increased in the other (B224) at the follow-up, while the maximal percent GzB release did not vary in either child, possibly limited by the intrinsic cellular capacity to release GzB. No clear conclusion can be drawn on the impact of ADCC on disease progression, due to the limited number of rapid progressors tested. However, in these children, the absolute titers were in the lower range (1/706 to 1/36,854) as compared to those of the slow progressing children (1/5860 to >1/156,250). While in slow progressing children ADCC and neutralization in terms of strength and breadth increased both at the late sample, in the two rapid progressing children this was not the case. At the late timepoint the plasma of child B380 showed a decrease of ADCC but an increase of neutralization breadth thought to low titers, and that of child B224 had an increased ADCC but neutralized only the Tier 1 SF162 PSV.

### 3.4. Lack of HIV-1 Neutralization Capacity Does Not Correlate with Impaired Antibody Responses in General

For the purpose of analyzing if the children displayed a generally altered ability to mount antibody responses, we tested their plasma against a spectrum of antigens, including an immunodominant HIV-1 gp41 epitope, JB-4, and childhood vaccine antigens, tetanus and diphtheria toxoids. Reactivity towards the JB4 peptide was observed in plasma from all 11 tested children and was maintained at similar levels independently from disease progression (Figure 4A). In most children the kinetics of the antibody response showed a drop with a subsequent increase within 3–9 months of age and persistence thereafter, indicative of a loss of maternally transferred antibodies and rise of the child’s own antibody production.

Analogously the antibody response towards tetanus and diphtheria toxoid was similar in rapid and slow progressors (Figure 4B,C). In general, the antibody titers to these antigens were higher during the first 18 months of age and eventually dropped thereafter, which clearly reflects the vaccination schedule. Thus, these data suggest that the children are capable of mounting immunodominant antibody responses independently of HIV-1 disease progression.

## 4. Discussion

In this study, we took advantage of a unique set of samples from HIV-1 infected children collected throughout disease progression in a prospective cohort study performed in the early years of the HIV/AIDS pediatric epidemy in Italy. We show that newborns displayed antibodies, which are transplacentally acquired, reactive with an immunodominant and constant HIV-1 gp41 epitope. However, neonate’s plasma usually did not neutralize the transmitted virus when tested in a PBMC-based assay. This suggests that the transmitted virus escapes neutralization already in the mother.

After the decline of maternal antibodies, the child’s own HIV-1 binding antibodies rose around 6 months and were followed by appearance of autologous Nabs within 12 months of age. These results are in line with a previous study showing that the development of autologous Nabs is rare before one year of age [36]. Another study described presence of autologous neutralization in four in utero infected children at three to four months of age [22], however, they made use of the TZMbl-assay, which may have a different sensitivity compared to the PBMC-based assay as we reported [46]. At difference, in infected adults, an autologous neutralizing response, detected with the PBMC-based assay, appears as early as within 4–8 weeks from infection [5,10]. It is difficult to ascertain if this difference is due to the intrinsic characteristics of the immune system of neonates, or the presence of antibodies of maternal origin, which may delay the development of the child’ own ones. These considerations may be of relevance for vaccination strategies designed to induce antibody responses in newborns. As today only one vaccine trial was performed in newborn administrating HIV-1 recombinant Canarypox vaccine and gp120 protein in a prime-boost regimen, which induced low titer anti-Env antibody responses [48].

We further show that slowly progressing children always develop a persistent and increasing autologous Nab response against the early virus, which instead is rare in rapid progressors (Figure 1). Furthermore, contemporaneous neutralization was rare and escape from neutralization was a common feature in all children, which indeed resemble very much findings described in HIV-1 infected adults [5,7,9].

Interestingly, we show that two-thirds of children, independently from disease progression, develop Nabs to Tier 2 viruses, AC10 and VI191, and almost 100% to Tier 1 SF162 within two years of age. Indeed, also Goo et al. showed that the majority of children developed cross-clade bNAbs by two years of age when tested with a TZMbl-assay [14], and Nabs’ breadth further correlated to viral load set point. Unfortunately, due to the lack of viral load data, we cannot conclude on its role. 

We observed a clear difference according to disease progression at the later timepoints assayed, by four years of age, when only slow progressors had a substantial increase in potency and breadth of their Nab response. In line with these results, we previously reported on the emergence of antigenically more diverse virus populations in slow than in rapid progressors of this same group of children, as a result of a stronger immune pressure [49]. Unfortunately, the study by Goo does not follow-up the children as long as we did, and thus does not allow for comparison.

The impaired Nab response in rapid progressors was not due to an impairment of the humoral immune response in general, as the children had antibody reactive to HIV-gp41 and the childhood vaccination antigens. However, Pensieroso et al. showed that B-cells are already perturbed early after HIV-1 infection in children, which in turn may contribute to less efficient antibody responses [50]. The administration of highly active ART translated into a better functional profile in that memory B-cell responses to HIV-1 and measles antigens were superior in early- compared with late-treated children, which indeed, was also shown for HIV-1 infected adults [51]. It is plausible to speculate that such B-cell impairment may affect also Nab responses, especially against escape virus variants, and accordingly merits further investigations.

Infected adults develop broad cross-Nabs by two to four years from infection only in 10–30% of cases [9,10,52,53]. Unfortunately, due to restrictions in blood drawings from the children we were limited to only three PSVs for testing heterologous Nabs with the TZMbl-assay. Interestingly, all of the children, who were infected with subtype B virus, still neutralized the subtype A Tier 2 VI191 virus at the late time point. Applying more stringent criteria, i.e., considering relevant only neutralization titers above 1/100, the frequency of children neutralizing all three PSVs dropped from 93% to 26%. In an Indian study, the frequency of inter-subtype cross-reactive Nabs in subtype C infected children, tested at the age of two and four years, varied, and ranged from 0 to 62% according to which of the three Tier 2 subtype B PSVs that was tested in the TZMbl-assay [12]. However, when we used PBMC-based assay and children’s primary isolates a heterologous Nab response was sporadically detected. The type of assay, PBMC- vs cell line-based, or primary virus isolate vs PSV are known to have different sensitivities and may provide different information [46]. 

We and others previously reported an association between pre-existing HIV-specific ADCC activity and better clinical outcome in infected children [11,18,20]. Here, our power to detect an impact of ADCC-mediating antibodies on disease progression in children was limited by the small number of rapid progressors analyzed. However, slow progressors showed consistently an increase in ADCC activity at follow-up. Although the observed trends support the hypothesis that Ab-mediated clearing of infected cells might slow down the disease, larger studies are needed to clarify if higher ADCC activity is a significant correlate of protection from rapid disease progression. It will also be of interest to assess if ADCC specific for the CD4-inducible epitopes, which has recently been suggested to be associated with decreased infant survival in a small cohort of infected children [54], is a trait of rapid progressors. Moreover, ADCC analyses using HIV-1-infected targets cells, displaying various Env epitopes, and the potential link to disease outcome in children would be interesting to explore.

## 5. Conclusions

In summary, our study showed that the autologous Nabs develop usually within 12 months of age to the transmitted/founder virus in slowly progressing HIV-1 infected children. These Nab responses persist throughout progression but induce continuous emergence of viral escape variants. Notably, despite rapid and slow progressor develop Nabs within two years from infection solely slow progressors show thereafter increase in potency and breadth of heterologous cross-Nabs within four years of age. Analogously antibody mediating ADCC developed within two years of age and increased thereafter in slow progressors. These findings suggest that immunocompetence impacts the ability to develop potent antibody reactivities in children. 

## Figures and Tables

**Figure 1 vaccines-09-00260-f001:**
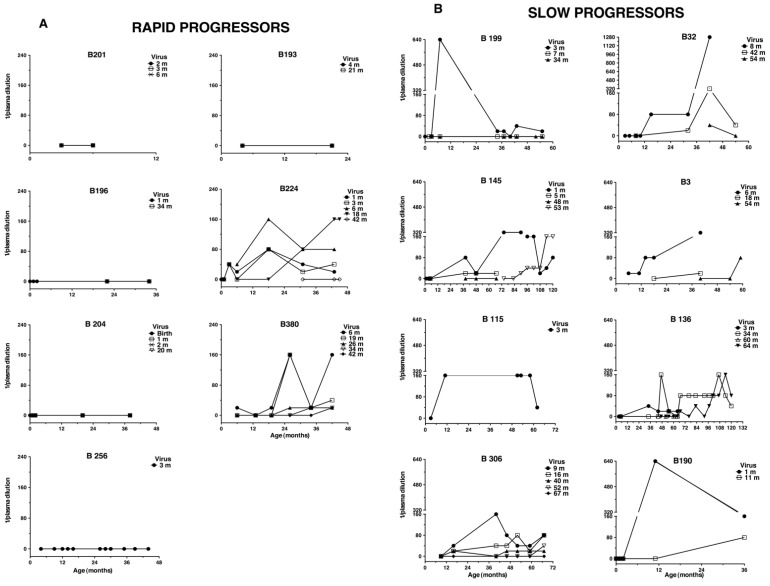
Autologous neutralization in PBMC-based assay of rapid (**A**) and slow progressing (**B**) children. Charts show the kinetic of neutralization of plasma obtained during disease progression against viral isolates from the same child. Neutralization is defined as the highest plasma dilution giving at least 90% reduction of viral production as detected by HIV-1 p24 antigen ELISA. 0 means <1/20 plasma dilution. Detection of autologous NAbs in slow progressors vs. rapid progressors was significant (*p* = 0.0070, Fishers’ exact test). “m” means month(s).

**Figure 2 vaccines-09-00260-f002:**
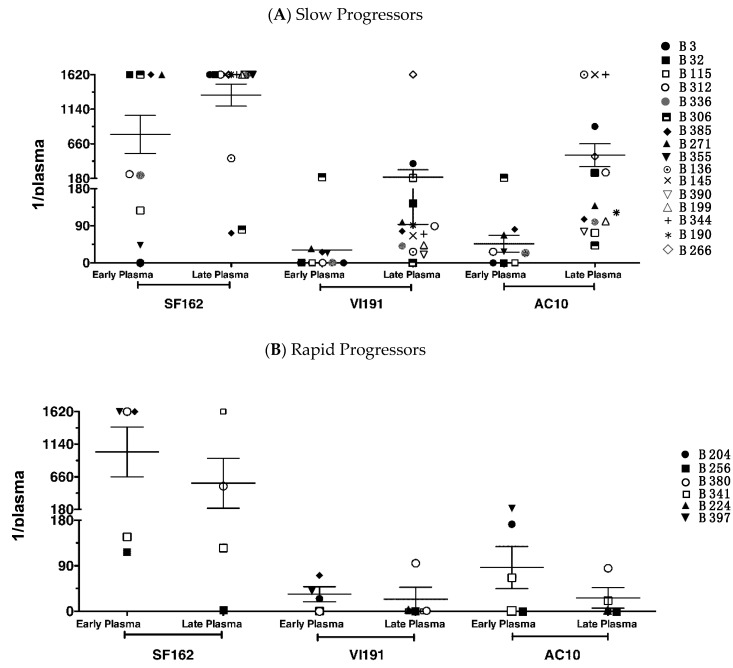
Neutralizing antibody response against heterologous PSVs in TZMbl-assay in children with slow (**A**) or rapid (**B**) disease progression. Heterologous Nab responses were tested with plasma of six rapid and 16 slow progressors, obtained within the second and/or fourth year of age. PSVs SF162 and AC10 are both subtype B and match the child’s infecting virus subtype, and PSV VI191 is subtype A. Indicated is the plasma dilution at which 50% inhibition was achieved. 1/1620 titer is used when an end-point titer was not achieved within the last plasma dilution of the test. Mean and SEM are also indicated. Comparison between groups of children showed a significant difference only for Nabs developed with late plasma against PSV AC10 (*p* = 0.0079, Mann-Whitney test).

**Figure 3 vaccines-09-00260-f003:**
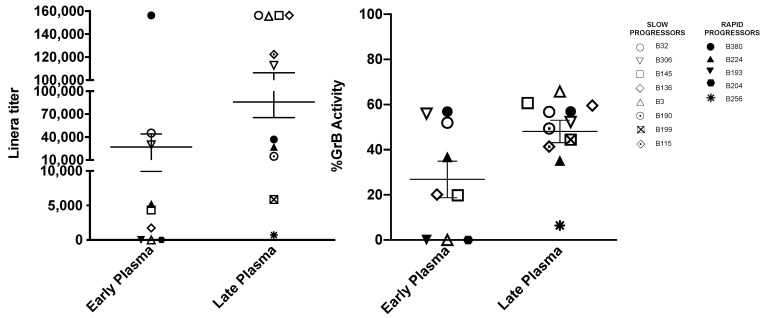
Child’s sero-reactivity to BALgp120 tested with ADCC assay. Indicated is the liner titer of the child’s plasma and the maximal %GrB release obtained at early and/or late timepoints during follow-up.

**Figure 4 vaccines-09-00260-f004:**
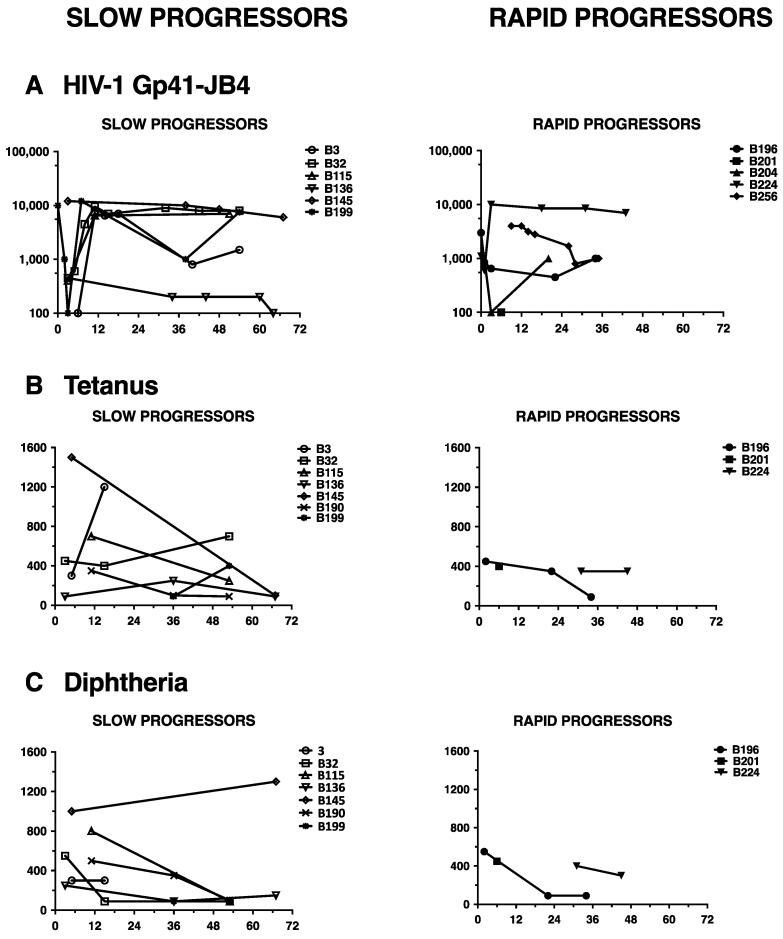
Reactivity to HIV-1 and childhood vaccine antigens in children with slow or rapid diseases progression. (**A**) to gp41 peptide JB4; (**B**) to tetanus toxoid; (**C**) to diphtheria toxoid.

**Table 1 vaccines-09-00260-t001:** Clinical, immunologic and virological characteristic of HIV-1 infected children ^(a)^.

Group of Disease Progression	Child Code	Age at First HIV Diagnosis ^(b)^	Age at Category Diagnosed ^(c)^			First CXCR4-Virus Isolation ^(d)^	Therapy Start ^(e)^	Death
			CDC 3	CDC B	CDC C			
Rapid Progressor								
	B193	4	6	6	12	21	12	28
	B196	1 (3d)	6	6	34	1	9	44
	B201	2	6	- ^(c)^	6	-	-	9
	B204	5d	6	6	-	1	-	38
	B224	1 (4d)	6	8	44	-	8	-
	B341	1	7	13	-	na	34	-
	B380	1	9	9	15	-	4	-
	B256	1.5	17	-	5	na	7	46
	B397	1	24	0.5	-	na	28	-
Slow Progressor								
	B32	3	28	28	-	-	28	96
	B199	1 (7d)	27	24	-	37	27	60
	B3	6	55	44	-	54	54	-
	B115	3	58	58	-	119	59	-
	B136	1	60	-	-	60	64	-
	B145	1	48	8	98	48	51	-
	B266	1	44	42	-	19	26	-
	B390	1	53	-	-	na	61	-
	B385	1	-	-	6	na	7	-
	B190	1 (3d)	-	-	-	-	77	-
	B271	1 (2d)	-	-	-	na	72	-
	B306	1	-	28	-	-	61	-
	B312	1	-	-	-	na	61	-
	B336	1	-	-	-	na	60	-
	B344	1	-	-	-	-	-	-
	B355	1	-	1.5	21.5	na	35	-

^(a)^ Age the event is occurring is indicated in months during a follow-up of 8 years. Symbol—means that the event has not occurred. Na means not available. D means days. ^(b)^ Age is indicated in months if not otherwise stated. HIV-1 diagnosis was performed with virus isolation and/or viral DNA PCR. In parenthesis is indicated the previous test which gave negative results, if performed. ^(c)^ Categories are defined according to the Centers for Disease Controls: CDC 3 = severe immune suppression; CDC B = moderate clinical manifestations; CDC C = severe clinical manifestations. Symbol—means that the event has not occurred. ^(d)^ Indicated is the child’s age when the first CXCR4-using virus was isolated, while previous viral isolates were R5. ^(e)^ Age of start of mono or dual antiretroviral therapy.

**Table 2 vaccines-09-00260-t002:** Heterologous neutralization of rapid and slow progressors against primary viral isolates tested in PBCM-based assay.

Plasma			Viral Isolate ^(a)^		
	Age ^(b)^	3–54 (X4) ^(c)^	32–42 (R5)	32–54 (R5)	145-5 (R5)
Rapid Progressor					
B193	24	0	40	nd ^(d)^	0
B224	31	0	0	0	0
B204	33	0	0	0	0
B256	39	nd	0	nd	0
Slow Progressor					
B199	37	0	0	0	0
B32	32	0	nd	nd	0
B145	38	80	0	0	(80) ^(e)^
B115	53	20	0	0	20
B136	34	0	40	20	0
B190	36	0	0	0	0

Neutralization titers are the reciprocal of the highest dilution giving a reduction of 90% of HIV p24 antigen. 0 means dilution < 1/20. ^(a)^ the virus isolate code (N-xx) is composed of the child’s identifier followed by the age in months of the child, from whom the virus was isolated; ^(b)^ Age in months; ^(c)^ Isolate phenotype; ^(d)^ “nd” means not done; ^(e)^ value in parenthesis indicate that the neutralization is against an autologous virus.

## Data Availability

The data presented in this study are available on request from the corresponding author.

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
