# Peer review of "Continuous HIV-1 Escape from Autologous Neutralization and Development of Cross-Reactive Antibody Responses Characterizes Slow Disease Progression of Children"

_vaccines, 2021, doi:10.3390/vaccines9030260_

Round 1
Reviewer 1 Report
Reviewer comments and suggestions
The current article demonstrated the kinetics of the autologous and heterologous neutralizing antibody (Nab) responses in HIV-1 infected children with different disease progression rates namely slow progressors and rapid progressors that monitored from after birth and five years continuous.
The method included 25 HIV-1 infected children included in this study were from another prospective study on MTCT conducted between 1989 and 1994. They quantified the autologous and heterologous neutralization measured by PBMC- and TZMbl-based assays, and cytotoxicity was assessed with the GranToxiLux assay.
The result included that newborns displayed antibodies directed towards the HIV-1 gp41 epitope, nevertheless, antibodies neutralizing the transmitted virus were undetectable.
The study noted that Nabs directed against the transmitted virus developed usually within one year of age in children with slow progression. Aautologous Nabs persisted throughout the follow-up of the slow progressors and induced the continuous emergence of escape variants.
Heterologous cross-Nabs were detected within two years, but their subsequent increase in potency and breadth was mainly a trait of slow progressors. The authors explained that kinetics responses to the immunodominant viral antigen and vaccine antigens were independent of disease progression.
The manuscript needs to be revised based on the below comments
- Second para of introduction ( third line ten should be 10)
- Same para of 5th line the author should discuss the different subtypes as they provided two references
- “disease progression in infected adults,10 suggesting that they do not play an essential role in preventing disease progression” please explore it and complete this sentence
- References cited 14-17 explore two to three-line about mechanism
- 5th para of introduction SHIV acquisition what does it mean
- (No need this line) and thus, more data are needed to understand how to proceed best for the prevention of MTCT, in specific MTCT through breastfeeding.
- “2 years of age, but the increase in potency and breadth thereafter is limited to slow progressors” need to cite an article for clearing the points
- “Children were all infected with a subtype B virus.38, 39 “As per the references I did not get any idea about stated sentences.
- Described previously better to explain little, not like previously described (I saw it numbers of time in manuscript)
- “Sequential plasma samples (between two and 16 samples/child) were obtained from 15 children, whose viral isolates were available, and were tested for autologous” three-time use of were not appropriate
- Better to present the sentences in a ray diagram for the feasibility of the reader (some sections of material and methods
- No discussion of ROC curve in statistic part
- Figure 2 and Figure 4 look blur, change with the original one
- “the maximal percent granzyme B release did not vary in either child”. What would be the probable reason for this
- You need to discuss this assay by comparing with the TZMbl and PBMC based assay
- “We further show that slowly progressing children always develop a persistent and increasing autologous Nab response against the early virus, which instead is rare in rapid progressors”. Mention the table number
- “cross-viral subtype Nabs in subtype C infected children, tested at the age of two and four years, varied, and ranged from 0” What type of assay they performed
- “Protection from rapid disease progression. It will also be of interest to assess if ADCC specific for the CD4-inducible epitopes:” It is better to include limitations of your study
- Check the references 39, 45,47,48
Author Response
We thank the reviewer for the comments and corrections to our manuscript, which have been very helpful. We have answered to each of those and changed the manuscript accordingly. we are confident that the manuscript has improved and read now better.
Reply to #1 reviewer’s comments and suggestions
The manuscript needs to be revised based on the below comments
- Second para of introduction (third line ten should be 10):
Reply: we corrected as suggested.
2. Same para of 5th line the author should discuss the different subtypes as they provided two references
Reply: the sentence has been clarified by assigning the correct reference to each statement.
3.“disease progression in infected adults,10 suggesting that they do not play an essential role in preventing disease progression” please explore it and complete this sentence
Reply: The sentence was changed. We think that now it is improved and the content clarified. The sentence now reads : “However, these cross-Nab responses are not necessarily associated with delayed disease progression in infected adults [10], suggesting that other virus controlling immune responses may play more essential roles in the determination of disease progression rates.”
4. References cited 14-17 explore two to three-line about mechanism.
Reply: we agree with the reviewer that it is important to provide some indications of potential mechanism and added the following sentence: “These papers proposed that exposure to high antigen concentrations at high CD4 T cell count as described in children, or the robust ability to mount responses mediated by T helper cells, may contribute to the development of bNAbs.”
5. 5th para of introduction SHIV acquisition what does it mean
Reply: We changed the sentence to clarify the meaning of SHIV acquisition: the sentence reads: “In this regard, it was clearly shown that cocktails of cross-neutralizing monoclonal antibodies (mAb) directed to HIV completely block acquisition of simian immunodeficiency virus expressing the HIV envelope (SHIV), in juvenile non-human primates when administrated before or even after oral administration of the virus [26,27], thus supporting the use of mAbs for prevention of infection.”
6. (No need this line) and thus, more data are needed to understand how to proceed best for the prevention of MTCT, in specific MTCT through breastfeeding.
Reply: We agree with the reviewer and have deleted the above statement.
7. “2 years of age, but the increase in potency and breadth thereafter is limited to slow progressors” need to cite an article for clearing the points.
Reply: We rephrased the sentence to clarify that these are our own results described in this manuscript.
8. “Children were all infected with a subtype B virus.38, 39 “As per the references I did not get any idea about stated sentences.
Reply: The references were changed. Indeed, reference 38 did not include any sequence data and subtype, while reference 39 and the new reference added, Scarlatti et al PNAS, have HIV envelope sequence data of the group of children included in the manuscript.
9. Described previously better to explain little, not like previously described (I saw it numbers of time in manuscript)
Reply: We changed the sentence and provided some details to clarify the method used to obtain the virus isolates from the patient’s peripheral blood mononuclear cells.
10. “Sequential plasma samples (between two and 16 samples/child) were obtained from 15 children, whose viral isolates were available, and were tested for autologous” three-time use of were not appropriate
Reply: The sentence was rephrased.
11. Better to present the sentences in a ray diagram for the feasibility of the reader (some sections of material and methods
Reply: We think that Supplementary table 1 contains the information of the assays performed for each child.
12. No discussion of ROC curve in statistic part.
Reply: The ROC curve stats is not applicable, and therefore not discussed in the Statistics Chapter.
13. Figure 2 and Figure 4 look blur, change with the original one
Reply: The original figures are provided.
14. “the maximal percent granzyme B release did not vary in either child”. What would be the probable reason for this.
Reply: The maximal granzyme B release may be limited by the intrinsic capacity of the cells to release and is not dependent from the dilution of the serum. We changed the sentence accordingly. In this same chapter we added an additional sentence to better explain and to compare the trend of ADCC and neutralization responses in the children during follow-up.
15. You need to discuss this assay by comparing with the TZMbl and PBMC based assay
Reply: We raised the issue of the difference between assays in the discussion in the second and seventh paragraphs. Indeed, we tested the sera of the children against primary autologous viral isolates in the PBMC-based assay and against heterologous pseudotyped viruses in the Tzmbl assay. From our experience of the NeutNet working group we know that neutralization titers are cell type and virus isolate dependent, and that a comparison with different assay but with the same virus provides different results which often do not correlate. Therefore, we prefer not to discuss such a comparison in the manuscript.
16. “We further show that slowly progressing children always develop a persistent and increasing autologous Nab response against the early virus, which instead is rare in rapid progressors”. Mention the table number
Reply: The figure number was added.
17. “cross-viral subtype Nabs in subtype C infected children, tested at the age of two and four years, varied, and ranged from 0” What type of assay they performed
Reply: the information was added in the sentence, which now reads: “In an Indian study, the frequency of inter-subtype cross-reactive Nabs in subtype C infected children, tested at the age of two and four years, varied, and ranged from 0 to 62% according to which of the three Tier 2 subtype B PSVs that was tested in the TZM-bl assay [12].”
18. “Protection from rapid disease progression. It will also be of interest to assess if ADCC specific for the CD4-inducible epitopes:” It is better to include limitations of your study
Reply: We discussed the issue and added the following: “Moreover, ADCC analyses using HIV-1-infected targets cells, displaying various Env epitopes, and the potential link to disease outcome in children would be interesting to explore.”
19. Check the references 39, 45,47,48
Reply: All references were checked.

Reviewer 2 Report
Dispenseri and colleagues performed a longitudinal study of two antiviral functions of antibodies – neutralization and antibody-dependent cellular cytotoxicity (ADCC)- in 25 HIV-1 infected infants. All children were infected with a subtype B strain. Nine were rapid progressors, as defined by clinical criteria. The major findings were (1) the absence of neutralization of autologous virus early after birth indicating infection with strains resistant to maternal antibodies; (2) slowly progressing children developed neutralizing antibodies more frequently than rapidly progressing children; (3) neutralization of contemporaneous autologous isolates was unfrequent, supporting the emergence of neutralization escape viruses. Using a pseudovirus-based neutralization assay the authors showed the increase of neutralization titers against heterologous strains overtime in children with slow disease progression. The authors showed the presence of antibody mediating ADCC. Some children without neutralizing antibodies had detectable antibodies againts vaccine antigens showing immune competence.
The strength of the work is the use of samples collected 25 to 30 years ago when antiretroviral therapy was not used for the prevention of HIV-1 transmission. Administration of one or two antiretroviral drugs to infants and children had no or modest effect on viral replication and antigenic stimulation of B cells. This cohort is exceptional to understand the development of HIV-specific response in newborns. The second strength is the use of virus isolated from 15 out 25 patients PBMCs, and the performance of neutralization assay using primary. The major limit is that the functional assays and testing of autologous viral strains have neither been performed for all children nor at all time points, due to sample availability. Overall the study is performed with current state-of-the art techniques, clearly presented and well discussed.
Minor comments
Is there any correlation between neutralization levels in PBMCS and TZMBLs assay?
Is there any correlation between ADCC titers and neutralizing antibody titers?
Figure 4, panels on rapid progressors, titers of anti-Tetanus and Diphteria: Symbols for child B201 appear as if it was part of child B196 follow-up.
Author Response
We thank the reviewer for the interest in our manuscript and the comments, which we have answered. We ade changes to the manuscript accordingly. We think that the manuscript has improved and now read well.
Reply to #2 reviewer’s Minor comments
Is there any correlation between neutralization levels in PBMCS and TZMBLs assay?
Reply: we tested the sera of the children against primary autologous viral isolates in the PBMC-based assay and against heterologous pseudotyped viruses in the Tzmbl assay. From our experience of the NeutNet working group and other collaborative efforts (see publications listed below) we know that neutralization titers of a given serum or monoclonal antibody are cell type and virus isolate dependent and that a comparison with different assays but with the same virus provides different results which often do not correlate. Therefore, we prefer not to discuss such a comparison in our manuscript. We agree, however, that this is an important point and raised the issue of the difference between assays in the discussion in the second and seventh paragraphs and refer to our previous works.
- L Heyndrickx, A Heath, E Sheik-Khalil, J Alcami, V Bongertz, M Jansson, M Malnati, D Montefiori, C Moog, L Morris, S Osmanov, V Polonis, M Ramaswamy, QJ Sattentau, M Tolazzi, H Schuitemaker, B Willems, T Wrin, EM Fenyö and G Scarlatti. International Network for Comparison of HIV Neutralization Assays: The NeutNet Report II. PLoSOne 2012, 7 (5): e364338. doi: 10.1371/journal.pone.0036438. Epub 2012 May 9.
- Fenyö EM, Heath A, Dispinseri S, Holmes H, Lusso P, Zolla-Pazner S, Donners H, Heyndrickx L, Alcami J, Bongertz V, Jassoy C, Malnati M, Montefiori D, Moog C, Morris L, Osmanov S, Polonis V, Sattentau Q, Schuitemaker H, Sutthent R, Wrin T, and Scarlatti G. International Network for Comparison of HIV Neutralization Assays: the NeutNet report. PLoS ONE 2009;4(2):e4505.
- V. Polonis, BK Brown, A Rosa Borges, S Zolla-Pazner, D S Dimitrov, M-Y Zhang, S W Barnett, R M Reprecht, G Scarlatti, E-M Fenyo, D C Montefiori, F E McCutchan, N L Michael. Recent Advances in the Characterization of HIV-1 Neutralization Assays for Standardized Evaluation of the Antibody Response to Infection and Vaccination. Virology 2008; 375(2): 315-20 [Epub ahead of print2008 Mar 24]
Is there any correlation between ADCC titers and neutralizing antibody titers?
Reply: The comparison of the titers of these two assays is not so obvious and direct, in sense that the neutralizing titer is an end-point serum dilution inducing the 50% neutralization of the virus growth, while the ADCC titer is the highest serum dilution inducing Granzym B release and calculated as the area under the curve. We therefore prefer not to compare the results as such. However, we agree that it is important, and therefore added a sentence in the Chapter on ADCC to better explain and to compare the trend of ADCC and neutralizing antibody responses in the children during follow-up.
Figure 4, panels on rapid progressors, titers of anti-Tetanus and Diphteria: Symbols for child B201 appear as if it was part of child B196 follow-up.
Reply: The figure was fixed.

Round 2
Reviewer 1 Report
The author has addressed all concerns.